# OpenReview forum: "Safe and Sound: Evaluating Language Models for Bias Mitigation and Understanding"
_NeurIPS.cc/2024/Workshop/SafeGenAi — SafeGenAi Poster_

### Official Review · Reviewer_4tVg · 2024-10-08
**Clear and complete experimental paper**

**Rating:** 8
**Confidence:** 5

**Review:**

Advantage points:
1. The figures and codes are concise and clear, with high readability
2. A large of detailed experiments, easy to reproduce
3. This paper fine-tuned the LLM on specific dataset, improving the model's ability to generate safe content while maintaining its language understanding capabilities.

Weakness:
1. Although it demonstrates its innovation in the application of existing technologies, this method is not a completely new idea.

---

### Official Review · Reviewer_ea89 · 2024-10-09
**Comment on "Safe and Sound: Evaluating Language Models for Bias Mitigation and Understanding"**

**Rating:** 6
**Confidence:** 4

**Review:**

This study introduces a specialized dataset consisting of unsafe content paired with benign variations, designed to train already-safe LLMs to further enhance their safety and responsibility while preserving knowledge integrity. The authors assess their instruction fine-tuning model on bias, toxicity, knowledge retention, and language understanding, demonstrating improved performance in these areas.

Strengths:
The introduction of a novel dataset for the development of safer and more responsible LLMs is a notable contribution. The authors effectively showcase the enhanced performance of their approach through a variety of evaluative metrics.

Limitations:
A key limitation lies in the use of a stronger model, GPT-4 Turbo, to generate debiased text for dataset construction. This reliance may present a bottleneck in the approach, especially if it fails to support weak-to-strong generalization. In such cases, the value of the study could be diminished.

Additionally, the authors have not compared their approach with previous methods cited in the literature review. This omission makes it difficult to clearly assess the comparative strengths and weaknesses of their method, leaving the significance of the contribution somewhat unclear.

---

### Official Review · Reviewer_VUVW · 2024-10-10
**Safe and Sound: Evaluating Language Models for Bias Mitigation and Understanding**

**Rating:** 4
**Confidence:** 4

**Review:**

Summary:

The paper proposes a method to obtain safe and unbiased outputs from an LLM without compromising with the language understanding capability. The key idea is to train a model on a self-curated dataset consisting of unsafe content paired with its safe alternative. The claim is that the resultant model is able to generate safe content without losing any of its credibility in understanding text. A number of experiments are conducted to demonstrate the LLM trained as per the proposed approach has less bias.

Strengths:

-	Overall, a large number of experiments are performed using different LLMs across multiple bias related metrics to show that the LLM trained as per the proposed approach outperforms zero-shot and few-shot prompted models.

-	The paper is generally written well, and therefore is easy to follow.

Weaknesses:

-	There is very limited novelty in any of the conducted experiments. The core thing seems to be that training on a combination of biased and unbiased dataset yields to a less biased model as compared to that trained on just a biased dataset. This really is not a surprising development.

-	Details regarding the tested dataset are missing, which makes it difficult to conclude anything regarding the experimental results. Moreover, parts of the dataset have been generated by an LLM (ChatGPT). It is unclear if observations made in this synthetic dataset can really scale to real-world applications.

-	[Minor] Looking at the large number of limitations mentioned in the paper, it seems like the authors have planned for a number of extensions to the submitted work. It is highly recommended that those experiments be completed, and the paper be resubmitted to fully highlight the insights discovered in the work.